# Cross-Cultural Adaptation and Validation of the Portuguese Version of the Metacognitive Prospective Memory Inventory—Short Form (MPMI-s) in Non-Central Nervous System Cancer Patients

**DOI:** 10.3390/healthcare13050463

**Published:** 2025-02-21

**Authors:** Filipa Santos, Ana Bártolo, Sara M. Fernandes, Ana F. Oliveira, Ana Paula Caetano, Isabel S. Silva, Jan Rummel, Pedro B. Albuquerque, Pedro F. S. Rodrigues

**Affiliations:** 1Department of Psychology and Education, Portucalense University, 4200-072 Porto, Portugal; 2CINTESIS.UPT@RISE-Health, Portucalense University, 4200-072 Porto, Portugal; 3INSIGHT: Piaget Research Center for Ecological Human Development, Piaget Institute—ISEIT/Viseu, 3515-776 Viseu, Portugal; 4Associação Fraunhofer Portugal Research—AICOS, 4200-135 Porto, Portugal; 5RISE-Health, Department of Education and Psychology, University of Aveiro, 3810-193 Aveiro, Portugal; 6Intrepid Lab, Lusófona University, 4000-098 Porto, Portugal; 7CLISSIS, Lusiada Research Center on Social Work and Social Intervention, 1349-001 Lisboa, Portugal; 8Department of Psychology, Heidelberg University, 69117 Heidelberg, Germany; 9CIPsi—Psychology Research Centre, School of Psychology, University of Minho, 4710-057 Braga, Portugal

**Keywords:** prospective memory, metacognition, compensatory strategies, oncology, validity, reliability

## Abstract

**Background/Objectives**: Cancer diagnosis and oncological treatments often lead to cognitive impairments, particularly in prospective memory, which affects the ability to recall future intentions. These difficulties can significantly impact therapeutic adherence, especially in the early stages of treatment, where timely medication and appointment adherence are critical. Despite this, effective measures for assessing prospective memory in cancer survivors remain limited. The current study aimed to translate and culturally adapt the Short Form of the Metacognitive Prospective Memory Inventory (MPMI-s) for use with Portuguese cancer survivors. **Methods**: The translation process involved back-translation, expert review, and pre-testing to ensure content validity. Psychometric evaluation was conducted with a sample of 111 cancer survivors [M(SD) = 49.3(9.4), ages 18–65], assessing internal consistency, factorial validity through principal components analysis, and convergent validity. **Results**: A final 18-item version of the MPMI-S demonstrated strong reliability and validity, comprising four factors: prospective forgetting, monitoring and planning strategies, imagery and visualization strategies, and external aid strategies. Significant correlations were found between these dimensions and traits such as conscientiousness, agreeableness, emotional stability, as well as distress and cognitive functioning. **Conclusions**: These findings underscore the potential of the MPMI-S as a valuable tool in clinical settings, offering insights not only into self-reported prospective memory abilities but also into the compensatory strategies employed by individuals in their daily routines. By integrating these aspects, this measure helps to identify key opportunities for rehabilitation aimed at minimizing the functional impact of the disease.

## 1. Introduction

Cancer is considered a significant public health issue in developed countries. This clinical condition represents a disruption in the life cycle due to its chronicity and short- and long-term side effects. Research has suggested the global impact of the disease, affecting individuals physically, emotionally, cognitively, and socially, from diagnosis through to survivorship [1]. Epidemiological data show that the number of new cancer cases has been increasing in recent years. In 2022, the global cancer incidence rate was 186.5/100,000 new cases, while in Portugal, the rate was 286.3/100,000. The most common types of cancer include breast, prostate, colorectal, trachea, bronchus, and lung cancers [2]. However, technological advances in early detection and treatments have changed the course of the disease, leading to higher survival rates and underscoring the need to invest in improving the quality of life for these patients.

In particular, the impact of cancer and cancer treatments on cognitive functioning is an increasingly recognized issue that has garnered the interest of the scientific community (internationally referred to as cancer-related cognitive impairment [CRCI]; [3]). Cognitive impairments may compromise survivors’ quality of life [4] and their ability to work, especially in working-age individuals [5]. The impacts of chemotherapy on cognitive function, commonly referred to as “chemobrain” or “chemofog”, demonstrate short-term consequences. However, evidence suggests that these effects may persist long-term, affecting 17% to 34% of patients [6,7,8,9,10].

The available literature, through neuropsychological testing, indicates deficits in various cognitive domains following cancer diagnosis, such as attention, memory, and executive functions [11]. However, this impact depends on the type of cancer and the treatment the individual undergoes, e.g., [6,12,13,14]. Further research has also shown that cancer patients often report significant cognitive complaints, which have been linked to psychological distress during the disease [15], particularly regarding prospective memory, e.g., [16,17].

In its broader sense, memory plays a fundamental role in human functioning. While it is commonly associated with past events, memory also involves the formation and execution of future actions. Prospective memory (PM) is one of the multiple memory systems that helps organize an individual’s functioning and is defined as the ability to plan an intention, retain it for a variable period, and retrieve it under appropriate circumstances [18,19]. Thus, failures in this domain can have serious consequences for individuals (e.g., forgetting to take medication at the prescribed time), making it essential to recognize patients’ primary complaints and the strategies they use to cope with potential impairments.

However, the availability of reliable and valid measures to assess prospective memory complaints remains a gap that impacts clinical intervention. Although the Prospective and Retrospective Memory Questionnaire [20] is available for the general population, it does not allow for the evaluation of the internal and external compensatory strategies used by individuals to manage their difficulties. Other available tools assess general cognitive impairment related to cancer, such as the Functional Assessment of Cancer Therapy-Cognitive Function Version 3 [21]. However, these tools do not clearly identify difficulties related to prospective memory, which are essential for determining the impacts in this context.

In this sense, the short version of the Metacognitive Prospective Memory Inventory (MPMI-s) emerges as a promising measure. Developed in Germany, it allows for a comprehensive assessment of perceived prospective memory capabilities, as well as the strategies used to cope with potential difficulties [22]. While originally developed for the general population, it integrates the common problems reported by cancer patients. Thus, the main objectives of the present study were as follows: (i) to translate and adapt the short version of the Metacognitive Prospective Memory Inventory (MPMI-s) to European Portuguese; and (ii) to explore the psychometric properties (reliability and construct validity) of the measure in a sample of cancer survivors.

## 2. Materials and Methods

### 2.1. Study Design and Participants

This study is part of a larger research project focused on individuals diagnosed with cancer at a young and active age. It is a validation, observational, and cross-sectional study that involves a convenience sample of cancer survivors. In the context of this study, the concept of “survivor” was broadly defined, considering all individuals from the moment of diagnosis, with or without evidence of disease in control exams at the time of recruitment [23]. Therefore, the inclusion criteria for participation in the study were as follows: (i) aged between 18 and 65 years; (ii) prior diagnosis of non-central nervous system (CNS) cancers; (iii) having undergone oncological treatments; and (iv) being a native Portuguese speaker. Participants with a history of brain metastasis and/or a diagnosis of dementia, epilepsy, brain injury, or substance abuse disorders were excluded from participation in the study.

### 2.2. Measures

The assessment protocol included a sociodemographic and clinical questionnaire to collect personal information such as age, marital status, education, employment status, psychiatric history, as well as information related to the disease such as cancer type, diagnosis date, method of disease detection, cancer stage, treatments received, pre-existing knowledge of memory difficulties, among others. The short version of the Metacognitive Prospective Memory Inventory (MPMI-s; [22]) in European Portuguese, translated and adapted by the research team throughout the present study, was administered, as well as measures to assess constructs that have been related to prospective memory, namely the Ten-Item Personality Inventory (TIPI; Portuguese Version: Nunes et al. [24]) for a personality assessment, the Hospital Anxiety and Depression Scale (HADS; Portuguese Version: Pais-Ribeiro et al. [25]) for psychological distress, and the European Organization for Research and Treatment of Cancer Quality of Life Questionnaire Core 30 (EORTC QLQ-C30; Portuguese Version: Pais-Ribeiro et al. [26]) for a quality of life assessment.

#### 2.2.1. Metacognitive Prospective Memory Inventory—Short Form (MPMI-s)

The short form of the MPMI was originally developed in Germany [22] to assess prospective memory and metacognitive processes in a general population sample. In its original version, it included three distinct scales: (a) Prospective Memory Ability (PMA), which measures the ability of individuals to remember their future intentions (e.g., fulfilling commitments); (b) Prospective Memory Strategy Internal (PMSi), which assesses how often participants use internal strategies (e.g., mentally visualizing future tasks); and (c) Prospective Memory Strategy External (PMSe), which measures the frequency of using external strategies (e.g., to-do lists). Responses were provided using a five-point Likert scale, ranging from 1 to 5: 1 = rarely, 2 = rather rarely, 3 = sometimes, 4 = rather often, and 5 = often.

##### Translation and Adaptation Process

In the present study, the original version of the MPMI-s was translated into European Portuguese after formal authorization from the original author [22]. Following best practices for developing and validating measures (Boateng et al. [27]), the translation was carried out independently by two bilingual translators fluent in German and Portuguese. One was a native German speaker with training in psychotherapy (T1), and the other was a translator with dual nationality (Portuguese and German) (T2). This approach ensured that both languages and cultural nuances were understood, and the terminology of the scale’s constructs was accurately preserved, ensuring that the items maintained the same meaning as the original language. Through this process, two translated versions of the MPMI-s were obtained (VT1 and VT2). Based on these two versions, the first reconciled version (VT3) was created and then back-translated. The reconciled version of the measure was analyzed by a panel of experts with Master’s (n = 1) and/or PhD (n = 4) degrees. The panel included two specialists in psycho-oncology and three in cognitive psychology, with a specific focus on human memory. The experts had the opportunity to conduct an independent review and suggested adjustments to the reconciled version of the MPMI-s. Semantic adaptations were made (related to phrasing, word accuracy, and meaning—e.g., “contract vs. signature”). Cultural equivalence was also considered, adapting terms based on cultural specifics (e.g., idiomatic expressions). All suggested changes were reviewed by the project coordination team (AB and PFSR) and incorporated into the measure. Finally, a pre-test was conducted with a sample of oncological patients (n = 3), all female, from different life stages (ages 25–62), with varying levels of education (from primary education to higher education), and a previous diagnosis of breast carcinoma (n = 2) or leukemia (n = 1), to assess the measure’s suitability for the context. Following a cognitive debriefing process, additional semantic adaptations were considered, focusing on word accuracy and meaning (e.g., ’post-its’ vs. ’sticky notes’), as well as language sensitivity to context (e.g., emphasizing the importance of personalized and ’close’ language in the instructions for completing the questionnaire). Figure 1 illustrates the different stages of the translation and adaptation process for this psychological assessment instrument (the European Portuguese version of the MPMI-s can be found in the Appendix A).

#### 2.2.2. Ten-Item Personality Inventory (TIPI)

The Ten-Item Personality Inventory (TIPI; Gosling et al. [28]; Portuguese version: Nunes et al. [24]) emerged from the need to quickly assess personality traits in many studies, leading to the development of brief scales (Nunes et al., 2018). This is a measure for assessing 5 personality traits based on the “Big Five” approach (extraversion, agreeableness, conscientiousness, emotional stability, and openness). It consists of 10 items that should be answered using a 7-point Likert scale (1—strongly disagree; 7—strongly agree). In the current sample, the reliability coefficient (Cronbach’s α) results were like those found in the validation study for the Portuguese population [24] and ranged from 0.32 to 0.63 for Openness to Experience and Extraversion, respectively.

#### 2.2.3. Hospital Anxiety and Depression Scale (HADS)

The 14-Item Hospital Anxiety and Depression Scale (Snaith & Zigmond [29]; Portuguese version: Pais-Ribeiro et al. [25]) is a tool that allows for the assessment of the severity of anxiety and depressive symptoms. It consists of 14 items that are evenly divided into two subscales, scored separately, assessing anxiety (HADS-A) and depression (HADS-D). Each item is answered using a 4-point Likert scale (from 0 to 3), and the total score for each subscale can range from 0 to 21 points. A higher final score indicates a greater presence of anxiety or depressive symptoms. This measure demonstrates good reliability [29]. It showed good internal consistency in the sample studied (α = 0.88 for anxiety; α = 0.80 for depression).

#### 2.2.4. European Organization for Research and Treatment of Cancer Quality of Life Questionnaire Core30 (QLQ-C30)

The European Organization for Research and Treatment of Cancer Quality of Life Questionnaire Core-30 (EORTC QLQ-C30) (Aaronson et al. [30]; Portuguese version: Pais-Ribeiro et al. [26]) is a self-report questionnaire designed to assess quality of life in individuals diagnosed with cancer. The use of this instrument is considered appropriate from the moment of diagnosis up to long-term survival. The questionnaire includes 30 items assessing five functioning scales (physical, role, cognitive, emotional, and social), three symptom scales (fatigue, pain, nausea and vomiting), one global health and quality of life scale, and single-item measures. Items are answered using a 4-point Likert scale, ranging from 1 (“Not at all”) to 4 (“Very much”), except for two items from the global health and quality of life scale, which use a modified 7-point scale (1—“Very poor” and 7—“Excellent”). The total score ranges from 0 to 100, as a linear transformation is used to standardize the raw score. A higher score on the functioning and general health perception items indicates better levels of functioning, while higher scores on the symptom scales reflect worse outcomes in terms of symptoms and difficulties [26,30]. In the present study, only the results from the five functioning scales were used. The internal consistency for the current sample was adequate for all subscales, ranging from 0.81 to 0.86 for physical and emotional functioning, respectively.

### 2.3. Ethical Considerations and Recruitment Process

This study was submitted for evaluation by an Independent Ethics Committee (P32–S52–10 May 2023). Participant recruitment occurred online through the dissemination of the questionnaire using the LimeSurvey platform hosted on the servers of the Universidade Portucalense. The study was promoted through patient associations in the community, as well as through pages and groups on various social media platforms (e.g., Facebook and Instagram). Prior to participating in the study, all participants had access to the informed consent form, and it was necessary for them to provide consent by selecting a checkbox. All ethical research procedures were followed in accordance with the Code of Ethics of Portuguese Psychologists and the Declaration of Helsinki, ensuring the anonymity and confidentiality of participants.

### 2.4. Data Analysis

Data analysis was conducted using IBM SPSS Statistics (version 29). Both descriptive and inferential analyses were performed. Frequencies, measures of central tendency, and dispersion were determined to characterize the sample in terms of sociodemographic and clinical characteristics and to explore the item properties of the MPMI-s. To assess internal consistency indicators, Cronbach’s alpha (α) and McDonald’s Omega (ω) were used. Although items 1, 4, 5, and 8 were related to prospective memory and, for this reason, were reversed in the original version, the guidelines of Rummel et al. [22] were followed. Only the inversion of items was considered for determining the final scores of the measure (Rummel et al., 2019). An exploratory factor analysis was also conducted using principal component analysis and the oblimin rotation method. The selection of exploratory factor analysis was made because this measure is originally developed and has not yet been validated for the oncology population. The convergent validity of the measure was explored by analyzing associations between the results of the MPMI-s and constructs such as personality traits, anxiety, depression, and quality of life, which have been associated with cognitive functions, particularly perceived prospective memory abilities, e.g., [15,31]. These analyses were conducted using Pearson’s correlation coefficient.

## 3. Results

### 3.1. Sociodemographic and Clinical Characterization of the Participants

The sample included 111 participants aged between 18 and 65 years (M = 49.3; SD = 9.4), of whom 96.4% (n = 107) were female and only 3.6% (n = 4) were male. Regarding the area of residence, approximately 52.2% of the participants lived in the northern region of Portugal (n = 58), 24.3% (n = 27) in the central region, 8.1% (n = 9) in the southern region, and 4.5% (n = 5) on the Azores islands. Most participants had completed an undergraduate degree (39.6%, n = 44), were married (57.7%, n = 64), and were employed full-time (57.7%, n = 64). The most frequently reported diagnosis was breast cancer (71.2%, n = 79), and, on average, participants had been diagnosed six years prior (M = 5.6; SD = 6.8; range 0–36). As for the most common oncological treatments, approximately 77.5% (n = 86) had undergone surgery, 73.9% (n = 82) had received chemotherapy, and 56.8% (n = 63) had undergone radiotherapy. However, 49.5% (n = 55) were still undergoing treatment, particularly hormone therapy (32.4%, n = 36). At the time of the study, 28.8% (n = 32) were receiving psychological or psychiatric support. Additionally, approximately 52.3% (n = 58) of the participants reported having other chronic comorbidities, with hypertension being the most frequent condition (15.3%, n = 17). The sociodemographic and clinical characteristics of the participants are summarized in Table 1.

### 3.2. Memory Difficulties and Pre-Existing Knowledge

Among the participants included in the study, 66.7% (n = 74) reported memory difficulties following their cancer diagnosis, particularly during (n = 25; 22.5%) and after chemotherapy (n = 21; 18.9%), as well as during endocrine therapy (n = 13; 11.7%). Approximately 15.3% (n = 17) could not specify when these difficulties emerged. On average, these difficulties lasted around 22 months (M = 22.3; SD = 24.8). Most survivors included in the study were unaware that cognitive difficulties could arise as a secondary symptom of cancer and/or its associated treatments (n = 59; 53.2%). Among the 46.8% (n = 52) of participants who had this knowledge, approximately 31.5% (n = 35) obtained the information from healthcare professionals, while 9.9% (n = 11) learned about it through support groups and/or other cancer patients.

### 3.3. Psychometric Properties of the MPMI-s

#### 3.3.1. Description of the Measure’s Items: Perceived Abilities and Strategies

A descriptive analysis of each item of the measure is presented in Table 2. The results indicate that the most frequently reported difficulties by participants were related to remembering tasks such as “Returning something someone lent me” (n = 44; 39.6%) and “Forgetting to call a friend back when I couldn’t reach them the first time” (n = 39; 35.1%). On the other hand, the data suggested that the least frequent difficulties (in the categories of “rarely or rather rarely”) were related to aspects such as “Canceling subscriptions in a timely manner” (n = 49; 44.1%), “Receiving payment reminders for forgetting to pay bills” (n = 40; 36%), and “Forgetting to send letters or emails on time” (n = 51; 45.9%). Regarding the most commonly used strategies to compensate for memory difficulties, the following stood out: verification (“I double-check if I’ve really done everything”; n = 65; 58.5%), creating lists (“I make a shopping list”; n = 67; 60.3%), planning (“I put something in my bag the night before so I don’t forget”; n = 90; 81%), and relying on stable routines and regularity (“Things I have to do regularly, I try to do at the same time every day”; n = 80; 72%).

#### 3.3.2. Item Properties

All possible response values on the Likert scale (1–5) were observed for each item. The median of most items was close to 3 (see Table 2). Table 3 shows that no deviations from normality were found, considering the absolute values of skewness (Sk < 3.0) and kurtosis (Ku < 7.0; [32,33]). All items showed significant corrected item-total correlations (r ≥ 0.35), except for items 2, 3, and 6, which relate, respectively, to task timing, returning borrowed items, and remembering phone calls. Item 3 exhibited the lowest correlation. These items also showed low inter-item correlations, mostly below 0.30. However, there was a slight variation in reliability when the items were excluded.

#### 3.3.3. Factorial Validity: Principal Component Analysis

The sample included in this study met the recommended 5:1 ratio for conducting an exploratory factor analysis of the measure’s factorial structure A principal component analysis with oblimin rotation was initially conducted, involving all the items. The Kaiser–Meyer–Olkin (KMO) measure showed a value of 0.87, and Bartlett’s test of sphericity was significant (ꭕ^2^(231) = 1050.527; *p* < 0.001), validating the correlation matrix structure. This analysis revealed five factors. However, the proposed structure was not conceptually justifiable since items 2 and 3 loaded separately onto a single factor. Therefore, based on the previously obtained item-total correlation results, items 2, 3, and 6 were excluded, and the analysis was re-conducted to test the factorial structure.

The KMO measure showed a value of 0.89, and Bartlett’s test of sphericity was also significant (ꭕ^2^(171) = 907.723; *p* < 0.001). Using the Kaiser criterion (eigenvalues > 1.00), the analysis identified four factors that accounted for 61.52% of the total variance. Interestingly, the resulting factor structure differs from the original grouping of strategies described by Rummel et al. [22], indicating possible variations in item clustering or conceptual organization within the dataset. Factor 1 (F1) included items 12, 18, 21, and 22, which were referred to in this study as “monitoring and planning strategies”. Factor 2 (F2) included items 9, 10, 11, 13, and 14, which related to internal strategies and were labeled as “imagery and mental visualization strategies”. Factor 3 (F3) grouped items 1, 4, 5, and 8, which were associated with “prospective forgetting”. Although item 15 showed a higher factorial loading on this factor, its loading was below 0.50 and did not conceptually fit the dimension assessed by the factor. This item also showed low communalities (≤0.5), justifying its exclusion from the measure’s structure. Finally, Factor 4 (F4) included items 7, 16, 17, 19, and 20, which assessed external recording strategies (such as lists and planners) and was labeled as “strategies related to the use of external aids”. For this factor, the inversion of item 7 should be considered, as it refers to the lack of using such external strategies. Additionally, Cronbach’s alpha coefficients for each subscale identified in the analysis ranged from 0.75 to 0.84, suggesting a good internal consistency for the new structure of the measure. Similar results were also found for the McDonald’s Omega, further supporting the reliability of the identified subscales (see Table 4).

#### 3.3.4. Convergent Validity: Relationship with Personality Traits, Distress and Quality of Life

The results suggested significant and positive intercorrelations between the four factors included in the measure. Regarding the relationship between the dimensions of the measure and other theoretically related constructs, it was found that difficulties reported in prospective memory (e.g., “prospective forgetting”) were negatively associated with personality traits such as conscientiousness (*r* = −0.225, *p* = 0.019). In contrast, the use of strategies related to external aids (e.g., lists and planners) was associated with agreeableness (*r* = 0.261, *p* = 0.006). Traits related to emotional stability were also positively correlated with the use of monitoring and planning strategies (*r* = 0.286, *p* = 0.003) and with the use of strategies related to external aids (*r* = 0.217, *p* = 0.024) to cope with prospective memory difficulties. Bivariate correlations with psychological distress showed that all dimensions of the MPMI-s were positively associated with anxiety. The strength of the association was moderate for most dimensions (0.403 ≤ *r* ≤ 0.456, *p* < 0.001), except for the use of the internal strategies of imagery and mental visualization, which showed a weak association (*r* = 0.262, *p* = 0.007). Similar results were observed for depressive symptoms, although no statistical significance was obtained for the association between depressive symptoms and internal strategies. Regarding the dimensions of quality of life, moderate associations were also found with the dimensions of the MPMI-s. All associations were negative, indicating that better emotional, physical, cognitive, role, and social functioning were associated with fewer difficulties in prospective memory, but also with a lower use of verification and planning strategies, as well as strategies related to the use of external aids (see Table 5). Cognitive functioning was the dimension that showed the strongest associations with all factors of the measure (−0.284 ≤ *r* ≤ −0.718, *p* < 0.001, for imagery and mental visualization strategies and prospective forgetting, respectively). Additionally, the correlation of the measure’s dimensions with the participants’ age was explored. However, the results showed only a weak negative association between strategies related to the use of external aids and age. Younger people seemed to use more external strategies of this nature (*r* = −0.239. *p* = 0.011).

## 4. Discussion

Prospective memory capabilities are crucial for daily activities, as well as for adherence to therapies and medical recommendations, particularly in the oncological context. This study aimed to translate, adapt, and assess the psychometric properties of the MPMI-s in a sample of Portuguese cancer survivors. The results suggest that the measure is reliable and valid for assessing perceptions of one’s prospective memory abilities, as well as the internal and external strategies employed to compensate for potential difficulties in this domain.

Based on the psychometric analysis, the final version of the MPMI-s in European Portuguese includes 18 items, presenting a structure that differs from the original version [22] but remains conceptually justifiable. The original scale consists of 22 items distributed across three distinct factors—PMA, PMSi, and PMSe. In contrast, the translated and adapted version for cancer survivors comprises a structure with four factors, offering a more detailed distinction between compensatory strategies by dividing them into monitoring and planning, imagery and mental visualization, and strategies involving referring to external aids. Notably, the factor of imagery and mental visualization stands out as an essential strategy for encoding and retrieving information. The literature supports this approach, showing that creating mental images facilitates memory consolidation and subsequent retrieval, e.g., [34]. Previous studies have emphasized that imagery-based strategies can assist individuals in overcoming cognitive difficulties, particularly those related to prospective memory [35]. This factor introduces a more advanced and practical approach for managing memory challenges, which is effectively reflected in the revised MPMI-s structure. Furthermore, the inclusion of a specific factor for “Prospective Forgetting” allows for a more direct assessment of the frequency and severity of perceived difficulties. In the original structure, this aspect was broadly incorporated into the first factor (PMA), whereas in an oncological sample—where such difficulties are often present [15]—it gains a more targeted focus, better addressing the needs and challenges of this population.

The results obtained with the MPMI-s indicate that the 18-item structure demonstrates good internal consistency, with Cronbach’s alpha coefficients ranging from 0.75 to 0.84. These values are consistent with those found in the original version [22], confirming the measure’s reliability in the Portuguese context. Moreover, other general tools, such as the FACT-Cog-v3 [21], have yielded similar reliability results in samples of oncology survivors, further highlighting the suitability of the MPMI-s for this population.

The MPMI-s also demonstrated strong indicators of convergent validity, reinforcing its usefulness in assessing prospective memory among cancer survivors. Research by McDaniel and Einstein [18] suggests that personality traits can influence performance on prospective memory tasks by affecting how intentions are retrieved and tasks are performed. In this study, analyses revealed a negative association between conscientiousness and “Prospective Forgetting,” aligning with findings that conscientious individuals—who tend to be organized, responsible, and careful—often employ planning and organizational strategies to complete daily tasks [36,37]. This behavioral consistency may create habits and routines that protect against prospective memory difficulties, making activities like adherence to therapeutic recommendations more automatic and less prone to being forgotten. Additionally, the use of strategies involving external aids, such as lists and agendas, was positively associated with agreeableness. The literature suggests that individuals with high agreeableness, who prioritize interpersonal relationships and social support, are more likely to adopt external strategies to manage their social and personal tasks, e.g., [38]. Emotional stability was also positively correlated with monitoring and planning strategies and external aids. These findings complement prior research suggesting that emotionally stable individuals are more likely to adopt methods that reduce the uncertainty and stress associated with forgetting. In contrast, individuals with higher levels of neuroticism tend to exhibit poorer performance on prospective memory tasks, as they are more susceptible to anxiety and stress, which can interfere with memory processes and hinder the use of effective coping strategies [39]. Despite the consistency of these results with the existing literature, the use of the TIPI to assess personality traits presented some limitations in terms of the reliability within this sample. While the reliability results were similar to those found in the Portuguese validation study [24], they were still considered somewhat weak. More robust instruments could provide more reliable and deeper insights into the personality traits of this population.

Nevertheless, distress symptoms, particularly depression, have consistently been recognized for their positive relationship with subjective memory complaints, as demonstrated by Rodrigues et al. [15]. Individuals with depressive symptoms tend to report greater cognitive difficulties, especially concerning memory, which may reflect the negative emotional impact on their ability to retain or recall information. The results of the present study are thus in line with the existing literature, which suggests that depression may exacerbate the perception of memory difficulties, influencing how people interpret and experience their memory failures. Moreover, the findings of this study support the association between anxiety and the use of compensatory strategies to manage difficulties related to future intentions. Anxiety, often characterized by excessive worry and emotional tension, can hinder the execution of tasks that require long-term planning and memory [40]. However, it is possible that anxious individuals resort to strategies such as using reminders or implementing structured plans to mitigate these difficulties and ensure that their intentions are not forgotten.

Another important indicator of convergent validity is the association between the MPMI-s dimensions and the cognitive functioning subscale of EORTC QLQ-C30, which has been used in various studies as a relevant measure of cognitive well-being in oncology populations [21]. The observed association suggests that the MPMI-s, by assessing aspects of prospective memory and compensatory strategies, is consistently related to global cognitive functioning and subjective perceptions of cognitive difficulties.

Interestingly, a generational trend was observed among younger participants, who demonstrated a greater tendency to adopt external compensatory strategies, such as calendars or lists. This generational trend was not observed in the original version of the MPMI-s [22], pointing to a shift in the approaches used to manage memory difficulties. This phenomenon may be related to the increased use of technology and digital tools for task and commitment management in everyday life, especially among younger generations. This finding underscores the importance of considering generational and cultural factors in psychometric evaluations, as these factors may influence the adoption and effectiveness of the cognitive strategies used to cope with memory difficulties.

By assessing prospective memory and the strategies used to compensate for cognitive challenges as proposed by the MPMI-s, this study provides valuable insights for future clinical interventions. Strategies such as monitoring and planning, imagery, and external aids can be integrated into cognitive rehabilitation programs to improve the quality of life of cancer survivors, based on a prior assessment of the difficulties reported by patients. The psychometric robustness of the European Portuguese version establishes its value as a practical and efficient tool for clinicians and researchers, demonstrating convergent validity by linking prospective memory with personality traits, emotional factors, and quality of life. Furthermore, evaluating perceived prospective memory abilities post-treatment through the administration of the MPMI-s could facilitate early intervention, with a more reduced impact on the individual’s functionality and on treatment adherence, which are often compromised by difficulties in this cognitive domain.

### Limitations and Future Studies

Despite the promising results obtained, several limitations warrant attention. First, it is important to note the homogeneity of the cancer types included in this study, which restricts the generalizability of the findings. Most participants had breast cancer (71.2%), although the MPMI-s is applicable to different types of cancer. Future studies should consider including a more diverse sample of cancer types to explore potential differences in prospective memory and the strategies use across various oncological conditions. Additionally, larger sample sizes are needed to test the invariance of the factorial structure obtained, particularly considering different therapeutic approaches (e.g., chemotherapy, endocrine therapy) and gender-specific variables. These aspects could not be adequately explored in the current study due to the limited number of participants in these subgroups.

It is also important to address the selection bias introduced by the online recruitment method. While this approach allowed for wider participation, it may have resulted in a sample that is not fully representative of the broader cancer patient population, particularly in terms of socio-demographic factors such as education level and geographical distribution. The sample was notably concentrated in the north of Portugal, and the educational background of the participants may have influenced the results. These factors should be carefully considered in future research to ensure a more balanced and diverse sample.

Regarding convergent validity, especially in relation to the personality traits assessment tool used, future research could benefit from incorporating other well-established instruments such as the NEO-PI-R [41] and the Big Five Inventory [42]. These tools have been previously applied in oncological populations and may provide a more comprehensive and reliable assessment of personality traits. Addressing the limitations outlined here—such as increasing the sample diversity and exploring additional variables and methodologies—will be crucial for consolidating the validity and applicability of the MPMI-s across different clinical and cultural contexts. This will, in turn, enhance the ability to assess and intervene in the cognitive functions affected by cancer, contributing to improved patient care and support.

## Figures and Tables

**Figure 1 healthcare-13-00463-f001:**
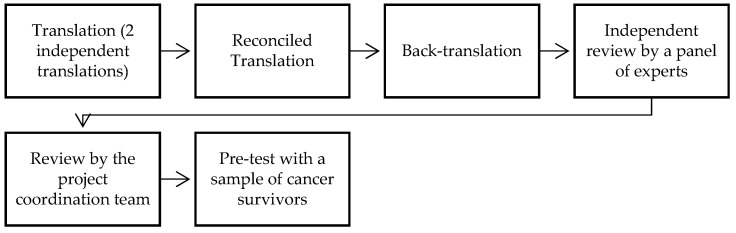
Steps of the process to translate and adapt the MPMI-s to European Portuguese.

**Table 1 healthcare-13-00463-t001:** Sample description.

Variable	n	%
Age in years (M ± DP; range)	49.32 ± 9.36; 23–65	
Education		
Elementary education	20	18
Secondary education	32	28.8
Undergraduate degree	44	39.6
Master’s degree	14	12.6
Marital status		
Single	15	13.5
Married	64	57.7
Divorced/separated	17	15.3
Cohabiting	13	11.7
Widower	2	1.8
Employment status		
Employed/Self-employed	71	64
Unemployed	9	8.1
Student	1	0.9
Disability pension	11	9.9
Sick leave	19	17.1
Time since initial diagnosis in years (M ± DP; range)	5.56 ± 6.78; 0–36	
Cancer type		
Breast	79	71.2
Uterus	3	2.7
Ovarian	2	1.8
Kidney	1	0.9
Colorectal	5	4.5
Lung	5	4.5
Non-Hodgkin’s Lymphoma	1	0.9
Hodgkin’s Lymphoma	1	0.9
Leukemia	2	1.8
Thyroid	7	6.3
Other	5	4.5
Previous treatments		
Surgery	86	77.5
Chemotherapy	82	73.9
Radiation therapy	66	56.8
Hormone therapy	53	47.7
Targeted therapy	6	5.4
Immunotherapy	16	14.4
Current stage of cancer treatment		
Undergoing treatment	55	49.5
Follow-up	56	50.5
Other chronic condition		
None	53	47.7
Hypertension	17	15.3
Diabetes	5	4.5
Kidney disease	0	0
Cardiovascular disease	0	0
Asthma	9	8.1
Other	27	24.3
Use of mental health services		
Yes	32	28.8
No	79	71.2

**Table 2 healthcare-13-00463-t002:** Descriptive analyses of the MPMI-s items: the frequencies, medians (Mdn), and interquartile ranges (IQR) of the MPMI-s.

Item *	Mdn (AIQ)	Rarely (%)	Rather Rarely (%)	Sometimes (%)	Rather Often (%)	Often (%)
1 _(PMA 1)_	3 (2)	32 (28.8)	17 (15.3)	35 (31.5)	22 (19.8)	5 (4.5)
2 _(PMA 2)_	3 (1)	11 (9.9)	12 (10.8)	43 (38.7)	28 (25.2)	17 (15.3)
3 _(PMA 3)_	3 (1)	7 (6.3)	19 (17.1)	41 (36.9)	28 (25.2)	16 (14.4)
4 _(PMA 4)_	3 (2)	13 (11.7)	19 (17.1)	40 (36)	29 (26.1)	10 (9)
5 _(PMA 5)_	2 (2)	40 (36)	25 (22.5)	29 (26.1)	13 (11.7)	4 (3.6)
6 _(PMA 6)_	3 (1)	11 (9.9)	16 (14.4)	41 (36.9)	32 (28.8)	11 (9.9)
7 _(PMA 7)_	3 (2)	16 (14.4)	28 (25.2)	31 (27.9)	22 (19.8)	14 (12.6)
8 _(PMA 8)_	3 (1)	26 (23.4)	25 (22.5)	43 (38.7)	13 (11.7)	4 (3.6)
9 _(PMSi 1)_	3 (1)	4 (3.6)	10 (9)	42 (37.8)	46 (41.4)	9 (8.1)
10 _(PMSi 2)_	4 (1)	10 (9)	16 (14.4)	22 (19.8)	46 (41.4)	17 (15.3)
11 _(PMSi 3)_	3 (2)	17 (15.3)	25 (22.5)	21 (18.9)	34 (30.6)	14 (12.6)
12 _(PMSi 4)_	4 (2)	11 (9.9)	12 (10.8)	23 (20.7)	36 (32.4)	29 (26.1)
13 _(PMSi 5)_	3 (1)	13 (11.7)	14 (12.6)	30 (27)	35 (31.5)	19 (17.1)
14 _(PMSi 6)_	3 (1)	9 (8.1)	11 (9.9)	45 (40.5)	34 (30.6)	12 (10.8)
15 _(PMSi 7)_	3 (3)	30 (27)	23 (20.7)	27 (24.3)	23 (20.7)	8 (7.2)
16 _(PMSe 1)_	4 (2)	14 (12.6)	12 (10.8)	24 (21.6)	31 (27.9)	30 (27)
17 _(PMSe 2)_	4 (3)	15 (13.5)	13 (11.7)	15 (13.5)	22 (19.8)	45 (40.5)
18 _(PMSe 3)_	4 (1)	3 (2.7)	3 (2.7)	15 (13.5)	36 (32.4)	54 (48.6)
19 _(PMSe 4)_	4 (3)	21 (18.9)	14 (12.6)	19 (17.1)	19 (17.1)	38 (34.2)
20 _(PMSe 5)_	4 (3)	18 (16.2)	16 (14.4)	21 (18.9)	25 (22.5)	31 (27.9)
21 _(PMSe 6)_	4 (2)	16 (14.4)	10 (9)	24 (21.6)	26 (23.4)	35 (31.5)
22 _(PMSe 7)_	4 (2)	4 (3.6)	9 (8.1)	18 (16.2)	41 (36.9)	39 (35.1)

* Note: The designation of each item on the original scale (Rummel et al., 2019 [22]) is inserted in parentheses to facilitate its identification.

**Table 3 healthcare-13-00463-t003:** Item properties: normality, item-total correlation, and internal consistency.

Item	Min–Max	Skewness	Kurtosis	Item-Total Correlation	Cronbach’s Alpha If Item Is Deleted
1	1–5	0.088	−1.111	0.432	0.833
2	1–5	−0.291	−0.429	0.238	0.840
3	1–5	−0.121	−0.538	−0.127	0.853
4	1–5	−0.188	−0.594	0.542	0.829
5	1–5	0.519	−0.722	0.354	0.836
6	1–5	−0.291	−0.435	−0.197	0.855
7	1–5	0.116	−0.929	−0.446	0.867
8	1–5	0.164	−0.611	0.396	0.834
9	1–5	−0.541	0.434	0.458	0.833
10	1–5	−0.580	−0.540	0.618	0.825
11	1–5	−0.129	−1.148	0.517	0.829
12	1–5	−0.616	−0.593	0.647	0.823
13	1–5	−0.410	−0.714	0.625	0.824
14	1–5	−0.400	−0.065	0.591	0.827
15	1–5	0.201	−1.112	0.468	0.831
16	1–5	−0.521	−0.838	0.572	0.826
17	1–5	−0.646	−1.007	0.504	0.829
18	1–5	−1.371	−1.825	0.509	0.831
19	1–5	−0.348	−1.349	0.547	0.827
20	1–5	−0.328	−1.217	0.590	0.825
21	1–5	−0.536	−0.929	0.623	0.823
22	1–5	−0.939	0.295	0.479	0.831

**Table 4 healthcare-13-00463-t004:** Principal component analysis: factor loadings, communalities, and internal consistency.

Items	F1	F2	F3	F4	H^2^
1	0.187	−0.038	**0.610**	−0.151	0.557
4	0.466	−0.058	**0.511**	−0.089	0.647
5	−0.224	0.126	**0.753**	−0.119	0.622
7	−0.190	0.271	−0.355	**0.531**	0.616
8	0.235	0.065	**0.674**	0.143	0.543
9	0.184	**0.636**	−0.303	−0.075	0.531
10	0.213	**0.762**	−0.034	0.015	0.697
11	−0.014	**0.763**	0.127	0.059	0.599
12	**0.516**	0.392	0.034	−0.090	0.618
13	−0.071	**0.664**	0.216	−0.223	0.659
14	−0.070	**0.772**	0.138	−0.092	0.672
15	0.002	0.337	0.450	−0.077	0.413
16	−0.043	0.097	0.059	**−0.823**	0.737
17	0.079	0.130	−0.212	**−0.741**	0.606
18	**0.748**	−0.004	−0.091	−0.171	0.656
19	−0.034	0.035	0.053	**−0.832**	0.718
20	0.453	−0.070	0.108	**−0.501**	0.682
21	**0.536**	0.068	0.313	−0.165	0.645
22	**0.631**	0.157	0.051	0.053	0.469
Mean (SD)	3.79 (0.91)	3.28(0.89)	2.58 (0.87)	3.37(1.09)	-
% Explained Variance	37.21	11.2	7.48	5.64	-
Cronbach’s Alpha (α)	0.77	0.84	0.75	0.84	_-
McDonald’s Omega (ω)	0.77	0.85	0.75	0.84	

Note: F1—Monitoring and planning strategies; F2—imagery and mental visualization strategies; F3—prospective forgetting; F4—strategies involving the use of external aids. Values in bold show the largest loading of each item in one factor.

**Table 5 healthcare-13-00463-t005:** Bivariate correlations.

	1.	2.	3.	4.	5.	6.	7.	8.	9.	10.	11.	12.	13.	14.	15.	16.
1. F1	1															
2. F2	0.573 ***	1														
3. F3	0.547 ***	0.358 ***	1													
4. F4	0.639 ***	0.407 ***	0.547 ***	1												
5. Extraversion	−0.016	0.004	0.027	0.011	1											
6. Agreeableness	0.145	0.084	0.056	0.261 **	−0.005	1										
7. Conscientiousness	−0.054	−0.112	−0.225 *	0.068	0.194 *	0.072	1									
8. Emotional Stability	0.286 **	0.140	0.144	0.217 *	−0.111	−0.240 *	−0.129	1								
9. Openness to Experience	0.044	0.043	0.129	0.052	0.509 ***	0.091	0.254 **	−0.179	1							
10. Anxiety	0.456 ***	0.262 ***	0.413 ***	0.403 ***	−0.155	0.006	−0.103	0.499 ***	−0.055	1						
11. Depression	0.335 ***	0.136	0.443 ***	0.239 *	−0.331 **	−0.188	−0.225 *	0.418 ***	−0.204 *	0.707 ***	1					
12. Emotional Functioning	−0.451 ***	−0.289 **	−0.442 ***	−0.414 ***	0.150	−0.027	0.040	−0.471 ***	−0.010	−0.808 ***	−0.614 ***	1				
13. Physical Functioning	−0.424 ***	−0.159	−0.406 ***	−0.302 **	0.028	0.099	0.104	−0.242 *	−0.079	−0.460 ***	−0.575 ***	0.440 ***	1			
14. Cognitive Functioning	−0.579 ***	−0.284 **	−0.718 ***	−0.635 ***	0.086	−0.004	0.132	−0.384 ***	−0.082	−0.584 ***	−0.554 ***	0.673 ***	0.476 ***	1		
15. Role Functioning	−0.278 **	−0.139	−0.378 ***	−0.240 *	−0.098	0.111	−0.151	−0.162	−0.289 **	−0.388 ***	−0.413 ***	0.472 ***	0.475 ***	0.401 ***	1	
16. Social Functioning	−0.340 ***	−0.096	−0.387 ***	−0.307 **	0.102	0.029	−0.070	−0.302 **	−0.184	−0.453 ***	−0.496 ***	0.526 ***	0.400 ***	0.453 ***	0.536 ***	1

Note: F1—Monitoring and planning strategies; F2—imagery and mental visualization strategies; F3—prospective forgetting; F4—strategies involving the use of external aids. * *p* < 0.05; ** *p* < 0.01; *** *p* < 0.001.

## Data Availability

The raw data supporting the conclusions of this article will be made available by the authors on request.

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
