# Peer review of "Cross-Cultural Adaptation and Validation of the Portuguese Version of the Metacognitive Prospective Memory Inventory—Short Form (MPMI-s) in Non-Central Nervous System Cancer Patients"

_healthcare, 2025, doi:10.3390/healthcare13050463_

Round 1

Reviewer 1 Report

Comments and Suggestions for Authors

The article describes the standardization of the Short Form of the Metacognitive Prospective Memory Inventory (MPMI-s) for cancer patients in the Portuguese language. The introduction is well articulated and the tools used in the research are well described. The meticulous procedure used to create the Portuguese version of the MPMI-s is completely convincing. One of the weak points, as underlined by the authors themselves, is the composition of the sample. If the total number is in itself satisfactory and allows for multivariate analyses, the inhomogeneities by type of tumor, course of the disease and type of therapy, resulting of the use of convenience sampling which suffers from self-selection due to the method of collecting the data, represents an objective problem.

However, the factorial structure of the instrument appears sufficiently robust and the relationships between the factors identified and the other measures used in the study are certainly interesting and in line with the expectations and data reported in the literature. However, the usefulness of the standardized instrument in Portuguese escapes the reader: what are the authors who commit to imagine for the MPMI-s? The type of relapse of the use of the tool on the quality of life of oncological patients should be deepened because the simple use for research purposes is not sufficient to justify such a timely process of standardization.

The conclusions that are currently not distinct compared to the discussion of the results should also be expanded. It does not seem sufficiently underlined that the causes of memory problems in oncological patients can be due on the one hand to psychological variables (anxiety, stress) on the other to the side effects of therapy.

A detailed observation: in the description of the European Organization for Research and Treatment of Cancer Quality of Life Questionnaire Core-30 (EORTC QLQ-C30) it is claimed to have a range of 0-100, but 28 items are rated on a scale of 1 to 4 and 2 on a scale from 1 to 7. Consequently, it is not clear how one can obtain a minimum score of 0 (the minimum should be 30) and a maximum score of 100 (if If you add the maximum values ​​you should have a maximum of 134 points). If there is an explanation (item to be subtracted) it would be useful to give it.

Author Response

[Manuscript ID: healthcare-3467253]

Dear Editor Dr. Jiang,

Thank you for the time spent reviewing our article and for the reviewers' valuable comments and suggestions. We have made an attempt to address the specific recommendations, to allow a final decision regarding the manuscript. Below, we present the list of adjustments that have been performed, following point-by-point the list of comments. All changes in the manuscript are highlighted in yellow.

We hope this new revised version is now suitable for publication and look forward to hearing from you.

Sincerely,

Pedro F. S. Rodrigues.

****

Review Report 1

The article describes the standardization of the Short Form of the Metacognitive Prospective Memory Inventory (MPMI-s) for cancer patients in the Portuguese language. The introduction is well articulated and the tools used in the research are well described. The meticulous procedure used to create the Portuguese version of the MPMI-s is completely convincing. One of the weak points, as underlined by the authors themselves, is the composition of the sample. If the total number is in itself satisfactory and allows for multivariate analyses, the inhomogeneities by type of tumor, course of the disease and type of therapy, resulting of the use of convenience sampling which suffers from self-selection due to the method of collecting the data, represents an objective problem. However, the factorial structure of the instrument appears sufficiently robust and the relationships between the factors identified and the other measures used in the study are certainly interesting and in line with the expectations and data reported in the literature.

A: We appreciate the reviewer’s important reflection. Indeed, despite the sample size, the results, both in terms of the reliability of the measure and its factorial and convergent validity, support the validity of the instrument. We acknowledge, and have pointed out in our limitations (page 15, line 469), the restriction in generalizing the results due to the homogeneity of the sample. Future studies should replicate our factorial structure to analyze the behavior of the measure across different types of cancer, therapeutic approaches, and even stages of the disease.

However, the usefulness of the standardized instrument in Portuguese escapes the reader: what are the authors who commit to imagine for the MPMI-s? The type of relapse of the use of the tool on the quality of life of oncological patients should be deepened because the simple use for research purposes is not sufficient to justify such a timely process of standardization.

A: We greatly appreciate the careful suggestion. Indeed, we attempted to demonstrate this early on in our introduction by justifying the relevance of validating the MPMI-s, in comparison to other instruments (page 2, line 70). However, in our discussion, this contribution may not have been emphasized enough. In this sense, we have made this idea clearer in the last paragraph of that section (page 14, line 456).

The conclusions that are currently not distinct compared to the discussion of the results should also be expanded. It does not seem sufficiently underlined that the causes of memory problems in oncological patients can be due on the one hand to psychological variables (anxiety, stress) on the other to the side effects of therapy.

A: We appreciate the suggestion. The last paragraph of the discussion, which broadly corresponds to the conclusion, has been expanded in line with the previous comment, highlighting the real clinical relevance of the measure. Regarding the causes of memory problems, we chose not to delve deeper into the discussion, as the main objective of this study was to assess the psychometric properties of the measure. However, we have included a paragraph that addresses this issue and incorporates data from a recent review conducted by our team (Rodrigues et al., 2023), which demonstrates the relationship between depressive symptoms and prospective memory complaints. This data was our primary focus of interest, as the associations found serve to assess the construct validity of the measure (page 14, line 427).

A detailed observation: in the description of the European Organization for Research and Treatment of Cancer Quality of Life Questionnaire Core-30 (EORTC QLQ-C30) it is claimed to have a range of 0-100, but 28 items are rated on a scale of 1 to 4 and 2 on a scale from 1 to 7. Consequently, it is not clear how one can obtain a minimum score of 0 (the minimum should be 30) and a maximum score of 100 (if If you add the maximum values ​​you should have a maximum of 134 points). If there is an explanation (item to be subtracted) it would be useful to give it.

A: Thank you for the detailed observation. As a matter of fact, the EORTC QLQ-C30 questionnaire contains 28 items rated on a scale from 1 to 4, and 2 items rated on a scale from 1 to 7. As mentioned, adding the maximum values of the items could result in a total score exceeding 100 points, which does not align with the final score range of 0 to 100. The scoring procedure for the EORTC QLQ-C30, as outlined in the manual, uses a linear transformation to ensure that final scores are within the range of 0 to 100. The process is as follows:

  • Calculation of the mean score: For each scale, the mean of the item scores is calculated. This value is called the raw score.
  • Linear transformation: This raw score is then transformed using a linear formula so that the final scores range from 0 to 100. In this transformation, a higher score represents a better level of functioning or a worse level of symptoms (depending on the scale).

For example, to ensure that the final scores fall between 0 and 100, an adjustment is made according to the maximum possible score and the actual value obtained. This means that, while the scores of each item may sum to more than 100, the linear transformation ensures that the final score is standardized to fall within the 0 to 100 range, maintaining consistency and comparability of results. The reference to the transformation process has been included in the manuscript (page 5, line 201).

Reviewer 2 Report

Comments and Suggestions for Authors

TitleCross-Cultural adaptation and validation of the Portuguese version of the Metacognitive Prospective Memory Inventory short form (MPMI-s) in non-CNS cancer patients 

Comments:

1. Is there any reported drug available to treat MPMI-s?

2. The non-CNS cancer patients in this study indicates which type of cancer? Authors need to include majorly in this study on the discussed cancer.

3. The sample size for analysing MPMP-1 in this study looks small. What’s the reason?

4. Is there any specific reason for selecting the participants in the northern and southern region?

Author Response

[Manuscript ID: healthcare-3467253]

Dear Editor Dr. Jiang,

Thank you for the time spent reviewing our article and for the reviewers' valuable comments and suggestions. We have made an attempt to address the specific recommendations, to allow a final decision regarding the manuscript. Below, we present the list of adjustments that have been performed, following point-by-point the list of comments. All changes in the manuscript are highlighted in yellow.

We hope this new revised version is now suitable for publication and look forward to hearing from you.

Sincerely,

Pedro F. S. Rodrigues.

***

Review Report 2

1. Is there any reported drug available to treat MPMI-s?

A. Considering the question raised, prospective memory difficulties have been effectively improved with Implementation Intentions interventions. This appears to be the most promising approach for intervening in this context. However, a review by Marton et al. (2019) indicates that this approach requires further study in the cancer. In order to promote the development of these approaches, it is crucial, first and foremost, to provide tools that assess the main difficulties perceived by these patients. The MPMI-s emerges as an alternative in this regard.

Marton, G., Bailo, L., & Pravettoni, G. (2021). Exploring the possible application of implementation intention on prospective memory of cancer patients. Cogent Psychology8(1). https://doi.org/10.1080/23311908.2021.1880303

2. The non-CNS cancer patients in this study indicates which type of cancer? Authors need to include majorly in this study on the discussed cancer.

A. Thank you very much for the reflection. However, the cancer types included in the study are described in Table 1 of the article. As mentioned in the “Limitations section” of the article, most of our participants had a breast cancer diagnosis. Therefore, future studies should consider the psychometric properties of the measure considering heterogeneous samples for a better understanding of the measure's behavior. Despite this, prospective memory complaints have been predominantly explored and reported in the context of breast cancer (e.g., Rodrigues et al., 2023).

Rodrigues, P. F. S., Bártolo, A., & Albuquerque, P. B. (2023). Memory Impairments and Wellbeing in Breast Cancer Patients: A Systematic Review. Journal of Clinical Medicine, 12(22), 6968. https://doi.org/10.3390/jcm12226968

3. The sample size for analysing MPMP-1 in this study looks small. What’s the reason?

A. Thank you very much! Indeed, the sample size could have been larger. However, considering that it was a clinical population and the recruitment was conducted online, there were some constraints in the recruitment process. Despite this, the total number is, in itself, satisfactory (5:1 ratio) and allows for multivariate analyses. Furthermore, the factor analysis is supported by the Kaiser-Meyer-Olkin (KMO) measure and Bartlett's test of sphericity.

4. Is there any specific reason for selecting the participants in the northern and southern region?

A. There is no specific reason for selecting participants from the northern and southern regions. This number arises from the fact that we had more healthcare institutions from these regions collaborating in the online dissemination of the study. The geographical distribution was, therefore, a consequence of the network of partnerships established for recruitment. The limitations section more clearly addresses the limitations related to the selection bias associated with conducting an online study (page 15, line 481).